# A Device Strategy-Matched Comparison Analysis among Different Intermacs Profiles: A Single Center Experience

**DOI:** 10.3390/jcm11164901

**Published:** 2022-08-20

**Authors:** Raphael Caraffa, Jonida Bejko, Massimiliano Carrozzini, Olimpia Bifulco, Vincenzo Tarzia, Giulia Lorenzoni, Daniele Bottigliengo, Dario Gregori, Chiara Castellani, Tomaso Bottio, Annalisa Angelini, Gino Gerosa

**Affiliations:** 1Cardiac Surgery Unit, Department of Cardiac, Thoracic, Vascular Sciences and Public Health, University of Padova, 35128 Padova, Italy; 2Unit of Biostatistics, Epidemilogy and Public Health, Department of Cardiac, Thoracic, Vascular Sciences and Public Health, University of Padova, 35128 Padova, Italy; 3Cardiovascular Pathology, Department of Cardiac, Thoracic, Vascular Sciences and Public Health, University of Padova, 35128 Padova, Italy

**Keywords:** mechanical circulatory support, left ventricular assist device, heart failure, device strategy

## Abstract

**Background****:** The present study evaluates outcomes of LVAD patients, taking into account the device strategy and the INTERMACS profile. **Methods:** We included 192 LVAD-patients implanted between January 2012 and May 2021. The primary and secondary end-points were survival and major adverse events between Profiles 1–3 vs. Profile 4, depending on implantation strategies (Bridge-to-transplant-BTT; Bridge-to-candidacy-BTC; Destination-Therapy-DT). **Results:** The overall survival was 67% (61–75) at 12 months and 61% (54–70) at 24 months. Profile 4 patients showed significantly higher survival (*p* = 0.018). Incidences of acute right-ventricular-failure (RVF) (*p* = 0.046), right-ventricular-assist-device (RVAD) implantation (*p* = 0.015), and continuous-venovenous-hemofiltration (CVVH) (*p* = 0.006) were higher in Profile 1–3 patients, as well as a longer intensive care unit stays (*p* = 0.050) and in-hospital-mortality (*p* = 0.012). Twelve-month and 24-month survival rates were higher in the BTT rather than in BTC (log-rank = 0.410; log-rank = 0.120) and in DT groups (log-rank = 0.046). In the BTT group, Profile 1–3 patients had a higher need for RVAD support (*p* = 0.042). **Conclusions:** LVAD implantation in elective patients was associated with better survival and lower complications incidence. LVAD implantation in BTC patients has to be considered before their conditions deteriorate. DT should be addressed to elective patients in order to guarantee acceptable results.

## 1. Introduction

End-stage heart failure (ESHF) is a major healthcare issue. Heart transplant (HT) represents the gold standard treatment, but it is limited by the shortage of available organs [1,2,3,4]. The results with LVAD implantation have significantly improved thanks to the advancement in technologies [5,6,7,8,9,10,11]. The enrolment of ambulatory patients in these programs is still under debate. In a recent prospective study of ambulatory patients with ESHF, LVAD therapy was associated with improved quality of life and patient’s functional capacity when compared with the optimal medical management [12]. The present study aims at evaluating the outcomes of LVAD patients, taking into account the device strategy and the Interagency Registry for Mechanically Assisted Circulatory Support (INTERMACS) profile.

## 2. Methods

### 2.1. Study Population

We designed a retrospective, observational, single center study including all ESHF patients who underwent LVAD implantation in our institution (University of Padua, Italy) between January 2012 and May 2021. Exclusion criteria were pediatric patients (younger than 18 years old), and the implantation of biventricular assist devices (BiVAD) or total artificial heart (TAH). Baseline demographics, clinical characteristics, laboratory data, echocardiography, and hemodynamic status were assessed at hospitalization. Intraoperative and postoperative data were prospectively collected, and patients were classified according to the INTERMACS profile definitions. We considered the device strategy assigned prior to implantation based on the intention to treat: bridge to transplantation (BTT), bridge to candidacy (BTC), and destination therapy (DT). Concerning the INTERMACS level, patients were assigned to two different groups: “critically ill” patients, including INTERMACS profiles 1–3, and “elective” patients, including the INTERMACS profile 4. All patients were followed up until death on device, orthotopic HT, or day of last observation, set on 30 April 2022.

### 2.2. Study Outcomes

The primary end-point of the study was overall survival. The secondary end-point was the onset of major adverse events.

### 2.3. Definitions

The INTERMACS Profile was defined according to the Interagency Registry for Mechanically Assisted Circulatory Support [6] and assigned to each patient by a dedicated team of cardiologists and cardiac surgeons before the implantation. The BTT strategy was based on the possibility of active insertion on the waiting list after the implantation. A patient was considered to be on a BTC when pulmonary resistances were high but reversible and represented a relative contraindication to transplantation. Other causes of BTC were malignancy on diagnostic definition, substance addictions, and obesity. In our cent, LVAD as a DT was reserved for patients eligible for mechanical support but too old to be transplanted (older than 70 years old) or with contraindication to transplantation, such as confirmed neoplasia.

Major outcomes were defined in accordance with the latest consensus statement on adverse events definitions of mechanical circulatory support [13]. Acute right ventricular failure (RVF) included early acute RVF and early post-implant RVF. Early acute RVF was defined by the need for implantation of a temporary or durable RVAD (including ECMO) concomitant with LVAD implantation (before the patient leaves the operating room). Early post-implant RVF was defined by the need for implantation of an RVAD within 30 days following implantation or failure to wean from inotropic support or inhaled nitric oxide within 14 days following implantation in the presence of clinical findings of RVF. Late RVF was represented by the need for implantation of an RVAD greater than 30 days after LVAD implantation or hospitalization greater than 30 days, which requires intravenous diuretic or inotropic support for at least 72 h and associated with RVF symptoms

### 2.4. LVAD Management

Indications for LVAD implantation followed the latest European Society of Cardiology, European Association for Cardio-Thoracic Surgery, and ISHLT guidelines [1,2,3,4]. All patients of our cohort were assisted with a continuous-flow generation LVAD. Devices used were: Jarvik 2000 FlowMaker™ VAD (Jarvik Heart Inc., New York, NY, USA), HeartWare HVAD™ System (Medtronic, Dublin, Ireland), and HeartMate 3™ LVAD (Abbott, Chicago, IL, USA). The implant was performed through a classic median longitudinal sternotomy or through minimally invasive approaches [14,15,16]. Intraoperative evaluation of the right ventricular function was crucial. Difficult weaning from cardiopulmonary bypass, hemodynamic compromise despite high doses of inotropes, and echocardiographic evidence of right ventricular distention recommended an RVAD implantation. Depending on the era, RVAD setting was central, using a Levitronix CentriMag (Levitronix LLC, Waltham, MA, USA) support, or percutaneous, TandemLife Protek Duo (TPD; TandemLife, Pittsburgh, PA, USA). At the end of the hospital stay, patients were referred to specialized cardiac rehabilitation units and then periodically evaluated in our dedicated outpatient clinic.

### 2.5. Statistical Analysis

Continuous variables are presented as median and interquartile range (IQR). The Student’s t-test for unpaired data or the Mann-Whitney U test was used to compare parametric and non-parametric continuous variables respectively (normal distribution was assessed by the Kolmogorov-Smirnov test). Categorical variables were presented as numbers and percentages. Χ^2^ analysis or the Fisher exact test was used to compare categorical variables where appropriate. A two-sided *p*-value of less than 0.05 was considered significant. Survival estimates were provided with the Kaplan–Meier method; data are represented as percentages and 95% confidence interval. Log-rank is reported and considered significant when less than 0.05. Patients were censored in case of HT or last follow-up available. A competing risk analysis was conducted in the BTT cohort to estimate the cumulative risk of a patient moving into one of three exclusive events: alive and on waiting list, HT, and death. All statistical analyses were performed using R software 3.3.5 [17] and SPSS software version 20 (IBM Corp., Armonk, NY, USA).

### 2.6. Study Oversight

Informed consent was obtained to participate in this study. Notably, the use of data for scientific and research purposes is already included in the surgical informed consent agreements. The local Institutional Review Board approved the study design, consent process, and review and analysis of the data (IBR number 48398/21). We also guarantee the respect of anonymity and professional secrecy and use the collected data and the statistical analysis solely for scientific purposes granted in accordance with the law in force (GDPR).

## 3. Results

### 3.1. Baseline Characteristics and LVAD Implantation

Table 1 summarizes all the baseline demographics and clinical characteristics, preoperative laboratory, and echocardiogram parameters. 192 patients underwent LVAD implantation for ESHF. The median age was 62 years (54–67), and 87% of them were males. Forty-five patients (23%) were classified as INTERMACS Profile 4. The most frequent device strategy adopted was BTT in 79 patients (41%), followed by BTC in 58 (30%) and DT in 55 (29%). Comparing the critically ill patients (147) to the elective patients (45), the sicker tended to be younger (*p* = 0.001) and more often required continuous dialysis treatment before implantation (*p* = 0.008). Moreover, their precarious hemodynamic condition was remarked by end-organ parameters such as higher BNP (*p* = 0.050), lower haemoglobin levels (*p* = 0.040) and sodium levels (*p* = 0.048), hepatic suffering with higher levels of AST (*p* = 0.002), ALT (*p* = 0.003) and bilirubin (*p* < 0.001), and higher C-reactive protein (*p* < 0.001). The data regarding LVAD implantation are collected in Table 2.

### 3.2. Overall Results

The pertinent data are represented in Table 3. Profiles 1–3 patients showed a higher incidence of acute right ventricular failure (RVF) (*p* = 0.046), RVAD implantation (*p* = 0.015), and acute kidney failure requiring dialysis (*p* = 0.006). The most frequent adverse event observed was bleeding (37%), requiring at least one surgical revision in almost all cases (85%), followed by cerebral events in 24% of the population, mainly ischemic (60%), with 40% mortality. Bowel complications were present in 13% of the patients, but less than 3% died due to intestinal ischemia.

In-hospital mortality of the overall LVAD population was 25%. The main causes of in-hospital death were multi-organ failures in 50%, with almost 70% of them caused by RVF. Neurologic events occurred in 25% of cases. The overall Kaplan–Meier survival was 68% (62–75) at 12 months and 63% (56–71) at 24 months (Figure 1). Stratifying patients according to the INTERMACS profile in two groups, Profile 1–3 versus Profile 4, the overall survival (Figure 2) was significantly higher in the latter (61% vs. 91% survival at 1-year, log-rank = 0.014).

### 3.3. Intention to Treat Analysis

#### 3.3.1. Survival

According to the intention-to-treat analysis (Figure 3), actuarial survival was considerably higher in BTT compared to other implantation strategies (log-rank < 0.001).

In the BTT group (Figure 4), 12-month and 24-month survival was 83% (73–95) and 78% (64–94) in Profile 1–3 patients and 95% (87–100) and 86% (68–100) in Profile 4 patients, respectively (log-rank = 0.390). A competing analysis was conducted in the BTT cohort to evaluate the cumulative risk of a patient being alive, HT, or dead (Figure 5). One year after the implantation, 40% of the patients were transplanted, and less than 10% died. These percentages increased, after two years, to more than 62% of patients receiving an HT and an overall mortality rate of 13%.

In the BTC group (Figure 6), after a higher initial perioperative mortality rate in Profile 1–3 patients, Kaplan–Meier analysis showed comparable survival (log-rank = 0.120). 12-month and 24-month survival rates were respectively 54% (42–70) and 51% (39–68) in Profiles 1–3 patients and 88% (67–100) and 88% (67–100) in patients with Profile 4.

In the DT group (Figure 7), 1-year and 2-year survival were respectively 41% (28–60) and 33% (21–53) in Profiles 1–3 and 88% (73–100) and 81% (61–100) in Profile 4 (log-rank = 0.046).

#### 3.3.2. Major Adverse Events

Major adverse events are reported in Table 3 In the BTT group, the need for RVAD support was higher in Profile 1–3 patients (*p* = 0.042). The incidence of other complications did not differ between groups. Only one death in the Profile 4 group was observed during the hospitalization (12% vs. 5%, *p* = 0.674) due to severe retroperitoneal hemorrhage and multi-organ failure.

In the BTC group, the in-hospital mortality rate in Profiles 1–3 was nearly quadrupled (44% vs. 12%, *p* = 0.129).

In the DT group, Profiles 1–3 patients showed a higher incidence of acute RVF (*p* = 0.005), RVAD implantation (*p* = 0.026), need for postoperative CVVH (*p* = 0.034), ICU length of stay (*p* = 0.050), and in-hospital mortality (*p* = 0.012).

## 4. Discussion

In order to understand the appropriate timing for LVAD implantation, several prospective studies compared patients with optimal medical management to those who received an LVAD [12,18,19].

The present investigation examines characteristics and outcomes from our single center LVAD experience. Based on different intention-to-treat (BTT, BTC, and DT) our cohort was stratified into two groups: “critically ill” and “elective” patients. The main findings were as follows:(a)among BTT patients a similar survival regardless of the INTERMACS profile was observed, though a higher incidence of RVAD implantation was seen in the critically ill group.(b)among BTC patients, slightly improved survival is seen in elective patients.(c)among DT patients, higher survival is evident for elective patients.

LVAD therapy has been established in recent years as the therapy of choice in ESHF [1,2,3,4]. The INTERMACS profile represents a fundamental prognostic tool to guide clinical decision-making. Patients in cardiogenic shock have worse outcomes after LVAD implantation [18,19,20]. A correct indication remains the key to success and improving patient selection allows us to maximize the benefits of therapy. In a recent prospective study [12] on ambulatory patients with ESHF (INTERMACS profile 4–7), LVAD therapy was associated with improved quality of life and functional capacity compared with continued optimal medical management. However, according to the last EUROMACS report, only 23% of implantations were performed in INTERMACS Profiles 4 or higher [8].

The risk-benefit assessment of LVAD therapy and its related adverse events guide the clinician’s orientation. The “frequent flyer” patient is hanging by a thread with his apparent but labile hemodynamic compensation, often being delayed in common clinical practice by the cardiologist towards advanced surgical therapies. The absence of a timely consideration for advanced surgical therapies may limit the maximum benefit achievable or be contraindicated at the time of evaluation. According to our analysis, comparable survival observed in BTT patients, regardless of the INTERMACS profiles, can guide the decision-making and patient orientation, justifying the delay in the treatment in light of a risk-benefit assessment that takes into account not only mortality but also adverse events and LVAD-related complications. Our evidence supports that good long-term survival can be achieved with LVAD therapy as BTT and represents a reasonable alternative for critically ill patients in an era with a shortage of donors. This reassures patients and their families, who are often concerned about not being on time anymore for LVAD support when they drop into lower INTERMACS profiles. In detail, our results showed that patients bridged to transplantation have a better survival rate than those with other strategies, with less than a 10% one-year mortality rate. Compared to EUROMACS [8] and INTERMACS registries [6,7,8] our results were similar. Additionally, in our experience a BTT patient has a 40% probability to be transplanted at 12 months, reaching over 62% at 24 months [6,8]. Limiting the waiting list time for transplantation results in a lower incidence of adverse events [21,22,23,24,25,26,27]. Our strategy allows this subgroup group of patients to benefit from excellent survival and to be in optimal condition for transplantation [25]. Although not significant, a trend towards a higher incidence of acute right ventricular dysfunction was noted in patients in INTERMACS Profile 1–3, with a greater proportion of patients requiring RVAD support. Despite this, in-hospital and overall survival were comparable.

As far as the BTC group is concerned, no differences in survival were observed. Unfortunately, the limited number of cases reduces the statistical power, and an adequate comparison is challenging. However, a slight trend towards better survival seems to be seen in elective patients. Some of these patients can be considered as a delayed referral with the progression of their ESHF that could have benefited from HT and/or LVAD implantation if evaluated in an earlier clinical phase. For this reason, advanced treatments should not be considered as the end point of therapeutic strategies when optimal medical management fails but as an integral part of the patient decision-making process.

Finally, DT patients showed a clearly different trend compared to the other groups. Preoperative unstable hemodynamic levels had an extremely poor prognosis in terms of complications and mortality. The rate of acute RVF was higher in the profile 1–3 patients, significantly contributing to the observed high in-hospital mortality. According to the latest INTERMACS registry, 47% of enrolled U.S. patients received an LVAD as DT treatment [6], and this percentage is expected to increase. We believe that this trend towards a broader DT indication is only admissible for elective patients.

These works provided key information on the risk-benefit balance to guide physician and patient orientation for elective LVAD therapy as a treatment of choice for ESHF. Our single center analysis aims to reinforce these considerations and support the clinician for the timely consideration of advanced treatments even in ambulatory patients.

## 5. Study Limitation

The main study limitation is its single center retrospective design. While this represents one of the greater LVAD series in Italy, as a single center, study the sample size may be limited. Despite our efforts to pursue LVAD therapy in elective patients, they often rejected the option of durable mechanical support in their early phase of disease and, as a result, the ambulatory group of our study was unbalanced. Moreover, by including more than ten years of experience with LVAD implantation, a learning curve has to be considered regarding patient management and postoperative management.

Three different devices were considered in our investigation. Although dissimilar rates of adverse events are described in the literature, no relevant differences in clinical outcomes were identified in our population in relation to the pump model. Finally, as we know that the implantation strategy is not an independent factor but relies on the clinical situation of the patient, we considered different INTERMACS profile in the same device strategy setting to limit this bias.

## 6. Conclusions

In conclusion, LVAD implantation in “elective” patients was associated with better survival and lower incidence of RVF and its complications. Our analysis showed a comparable survival in “critically ill” and “elective” BTT, while a significantly higher mortality rate was observed for “critically ill” DT patients. LVAD implantation in BTC patients must be considered before their conditions deteriorate.

## Figures and Tables

**Figure 1 jcm-11-04901-f001:**
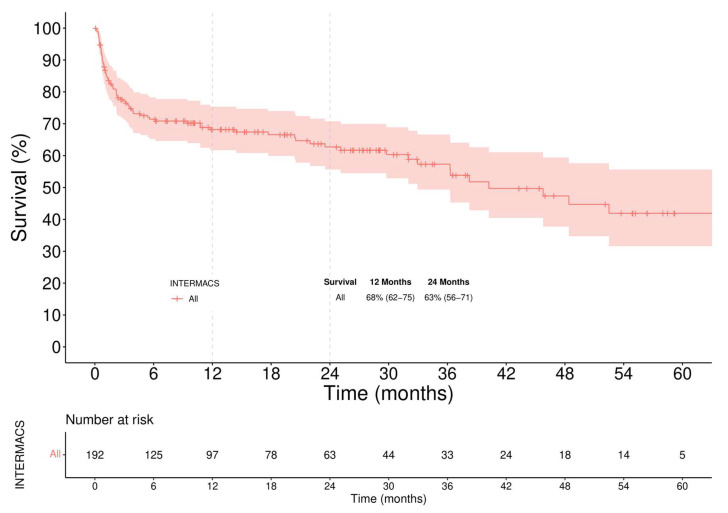
Overall Survival. Patients were censored at time of transplant or last follow-up.

**Figure 2 jcm-11-04901-f002:**
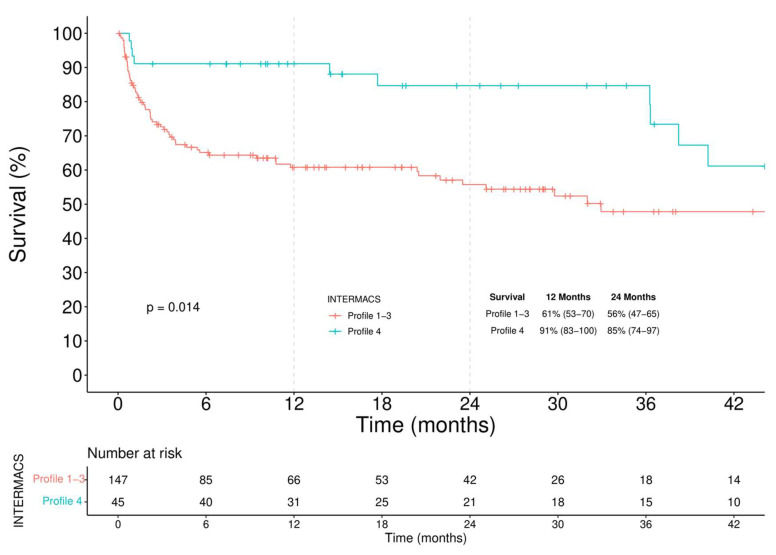
Kaplan-Meier analysis of survival stratified by baseline INTERMACS patient profile.

**Figure 3 jcm-11-04901-f003:**
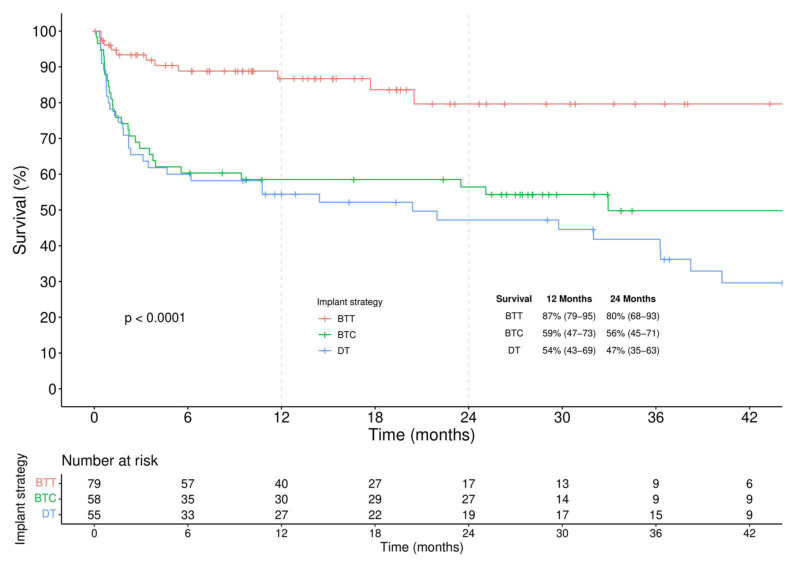
Kaplan-Meier analysis of survival stratified by different implantation strategies.

**Figure 4 jcm-11-04901-f004:**
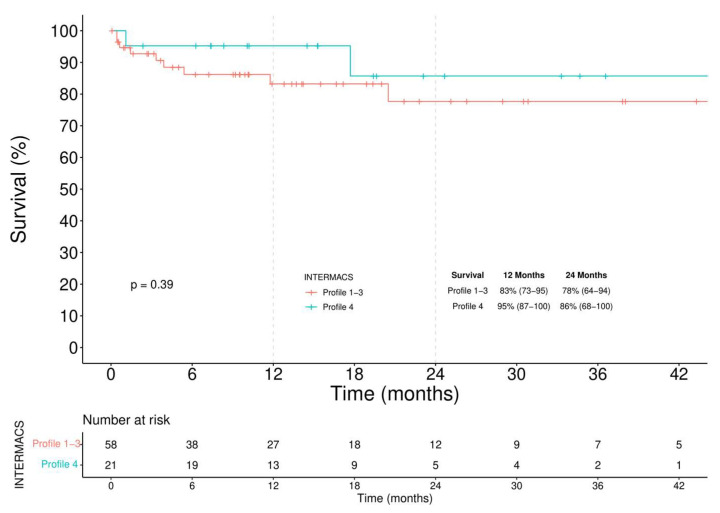
Kaplan–Meier analysis of survival among INTERMACS Profile 1–3 and Profile 4 BTT patients.

**Figure 5 jcm-11-04901-f005:**
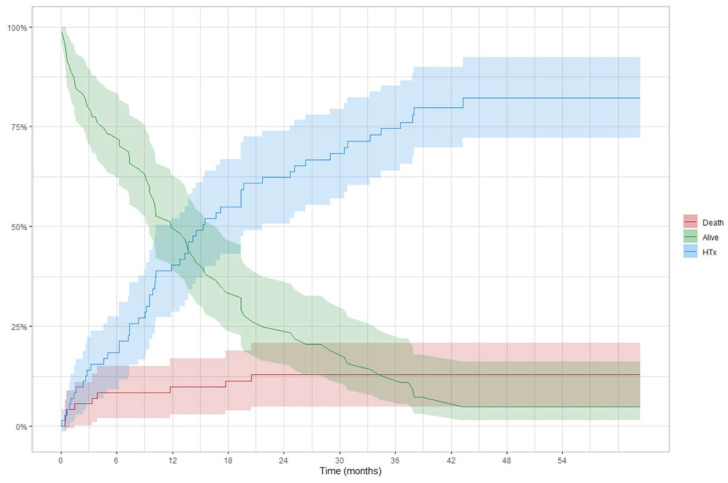
Competing risk analysis in BTT patients: curves are used to represent the cumulative risk of a patient during the time of being alive, transplanted, or dead.

**Figure 6 jcm-11-04901-f006:**
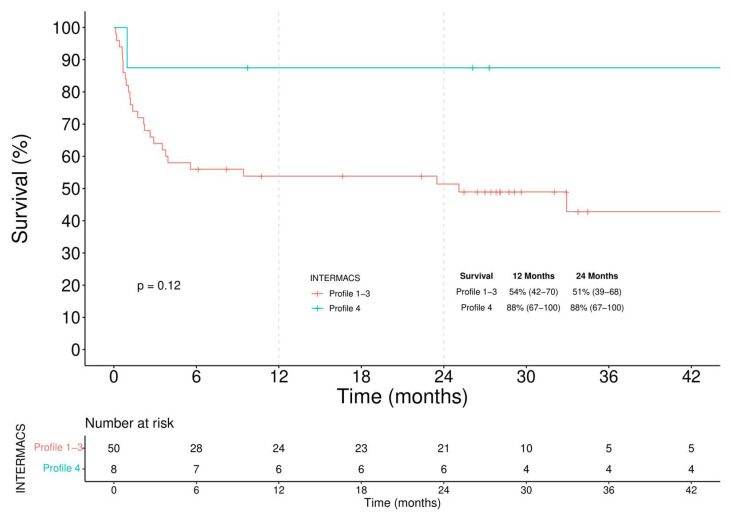
Kaplan–Meier analysis of survival among INTERMACS Profile 1–3 and Profile 4 BTC patients.

**Figure 7 jcm-11-04901-f007:**
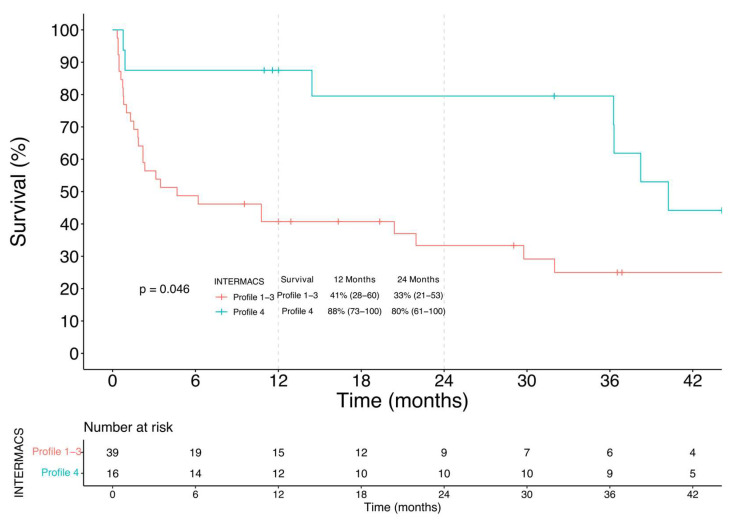
Kaplan-Meier analysis of survival among INTERMACS Profile 1–3 and Profile 4 DT patients.

**Table 1 jcm-11-04901-t001:** Baseline characteristics, preoperative labs, and echocardiogram parameters of overall (*n* = 192), INTERMACS Profile 1–3 (*n* = 147) and Profile 4 (*n* = 45) patients.

Baseline Characteristic	Overall (*n* = 192)	Profile 1–3 (*n* = 147)	Profile 4 (*n* = 45)	*p*
Age at Implant	62.4 (54.5–66.9)	60.6 (52.6–66.2)	64.9 (61.3–68.7)	0.001
Gender (male)	167 (87%)	124 (84.4%)	43 (95.6%)	0.073
Cardiac Diagnosis				0.389
DCM	86 (44.8%)	69 (49.6%)	17 (37.8%)	
IHD	94 (49.0%)	68 (46.3%)	26 (57.8%)	
Other	12 (6.2%)	10 (6.8%)	2 (4.4%)	
INTERMACS class				
1	60 (31.2%)			
2	27 (14.1%)			
3	60 (31.2%)			
4	45 (23.4%)			
Device Strategy				0.898
Bridge to transplantation (BTT)	79 (41.1%)	58 (39.5%)	21 (46.7%)	
Bridge to candidacy (BTC)	58 (30.2%)	50 (34.0%)	8 (17.7%)	
Destination therapy (DT)	55 (28.7%)	39 (26.5%)	16 (35.6%)	
Type of device				0.528
Jarvik	62 (32.3%)	45 (30.6%)	17 (37.8%)	
HVAD	54 (28.1%)	44 (29.9%)	10 (22.2%)	
Heart Mate 3	76 (39.6%)	58 (39.5%)	18 (40.0%)	
Smoker	80 (41.7%)	61 (41.5%)	19 (42.2%)	0.931
Dyslipidaemia	95 (49.5%)	66 (44.9%)	29 (64.4%)	0.027
Hypertension	109 (56.8%)	78 (53.1%)	31 (68.9%)	0.085
AF preop	73 (38.0%)	56 (31.3%)	17 (37.8%)	0.969
Cancer	17 (8.9%)	14 (9.5%)	3 (6.7%)	0.766
Diabetes	58 (30.2%)	43 (29.3%)	15 (33.3%)	0.584
Peripheral vascular disease	50 (26.0%)	32 (21.8%)	18 (40.0%)	0.020
COPD	17 (8.9%)	13 (8.8%)	4 (8.9%)	0.993
BSA (m^2^)	1.88 (1.77–2.00)	1.88 (1.74–2.00)	1.90 (1.80–1.99)	0.573
ICD	129 (67.2%)	88 (59.9%)	41 (91.1%)	<0.001
Redo	30 (15.6%)	20 (13.6%)	10 (22.2%)	0.167
Cardiac Index (L/min/m^2^)	2.00 (1.95–2.05)	1.98 (1.95–2.00)	2.00 (1.92–2.05)	0.040
Preop vO2 Peak (mL/min/m^2^)	11.0 (9.0–12.0)	11.0 (9.0–12.0)	11.0 (10.5–12.5)	0.136
Preop CVVH	18 (9.4%)	18 (12.2%)	0 (0.0%)	0.008
Preop CVVH days	5.5 (3.8–8.5)	5.5 (3.8–8.5)		
Preop ECMO	48 (25.0%)			
Days	5.0 (2.0–10.0)			
Preop temporary-LVAD	27 (14.1%)			
Days	7.0 (4.0–11.0)			
Preop Bi-VAD	2 (1.0%)			
Days	12.0 (10.0–14.0)			
Preop Intubation	35 (18.2%)	35 (23.8%)	0 (0.0%)	<0.001
Days	3.5 (1.0–6.0)			
**Preoperative Labs**	**Overall (*n* = 192)**	**Profile 1–3 (*n* = 147)**	**Profile 4 (*n* = 45)**	** *p* **
Haemoglobin g/L	11.0 (9.6–12.6)	10.6 (9.4–12.3)	12.1 (10.6–13.5)	0.040
Platelets count 10^9^/L	209 (155–269)	211 (155–278)	201 (159–239)	0.759
D-dimer	664 (277–1914)	824 (310–2322)	301 (202–552)	0.010
Bilirubin (mg/dL)	0.95 (0.58–1.60)	1.05 (0.64–1.89)	0.74 (0.45–1.13)	<0.001
AST (U/L)	28 (20–42)	30 (21–46)	22 (16–30)	0.002
ALT (U/L)	25 (17–41)	26 (18–46)	22 (15–35)	0.007
Amilasi (U/L)	28 (19–60)	30 (20–67)	25 (16–36)	0.001
LAD (U/L)	267 (192–421)	287 (211–450)	208 (160–273)	0.057
CRP (mg/L)	19 (4–84)	32 (13–97)	3 (3–11)	<0.001
Na (mmol/L)	137 (134–140)	137 (134–139)	140 (136–142)	0.048
BNP	6267 (3555–12,460)	8233 (3794–13,874)	3818 (1199–7560)	0.050
GFR (ml/min/m^2^)	64.0 (45.0–90.0)	68.0 (49.0–90.0)	56.0 (43.0–79.0)	0.182
Creatinine (mg/dL)	1.26 (0.94–1.59)	1.24 (0.90–1.56)	1.39 (1.03–1.61)	0.567
**Preoperative Echocardiogram**	**Overall (*n* = 192)**	**Profile 1–3 (*n* = 147)**	**Profile 4 (*n* = 45)**	** *p* **
EF (%)	20 (17–26)	20 (17–24)	25 (20–29)	0.001
iLVEDV (mL/m^2^)	130 (110–157)	129 (106–157)	135 (114–162)	0.373
TAPSE (mm)	15.0 (13.0–19.0)	15.0 (12.0–18.1)	18.2 (14.3–20.8)	0.771
sPAP (mmHg)	43 (35–54)	43 (36–53)	44 (32–44)	0.758
RV FAC (%)	31 (24–39)	29 (22–38)	36 (29–40)	0.002
TR ≥ moderate	82 (42.7%)	64 (43.5%)	18 (40.0%)	0.732
MR ≥ moderate	112 (58.3%)	87 (59.2%)	25 (55.6%)	0.731
AR ≥ moderate	4 (2.1%)	3 (2.0%)	1 (2.2%)	0.941

Abbreviations: ALT, alanine transaminase; AR, aortic regurgitation; AST, aspartate transaminase; Bi-VAD, bi-ventricular assist device; BNP, B-type natriuretic peptide; BSA, body surface area; BTC, bridge to candidacy; BTT, bridge to transplantation; CRP, C-reactive protein; CVVH, continuous venovenous hemofiltration; DCM, dilated cardiomyopathy; DT, destination therapy; ECMO, extrac corporeal membrane oxygenation; EF, ejection fraction; GFR, glomerular filtration rate; ICD, implantable cardioverter defibrillator; IHD, ischemic cardiomyopathy; iLVEDV, indexed left ventricular end-diastolic volume; LDH, lactate dehydrogenase; LVAD, left ventricular assist device; MR, mitral regurgitation; RV FAC, right ventricle fractional area change; sPAP, systolic pulmonary artery pressure; TAPSE, tricuspid annular plane systolic excursion; TR, tricuspid regurgitation.

**Table 2 jcm-11-04901-t002:** LVAD implantation characteristics of overall (*n* = 192), INTERMACS Profile 1–3 (*n* = 147) and Profile 4 (*n* = 45) patients.

LVAD Implantation	Overall (*n* = 192)	Profile 1–3 (*n* = 147)	Profile 4 (*n* = 45)	*p*
Surgical access				
Full sternotomy	110 (57.3%)	87 (59.2%)	23 (51.1%)	0.338
Minimally invasive	82 (42.7%)	60 (40.8%)	22 (48.9%)	0.390
Outflow graft anastomosis				
Ascending aorta	154 (80.2%)	122 (83.0%)	32 (71.1%)	0.090
Descending aorta	27 (14.1%)	19 (12.9%)	8 (17.8%)	0.463
Left subclavian artery	11 (5.7%)	6 (4.1%)	5 (11.1%)	0.134
Associated procedures	25 (13.0%)	16 (10.9%)	9 (20.0%)	0.130
Device Implantation				
CPB	121 (63.0%)	90 (61.2%)	31 (68.9%)	0.383
ECMO	31 (16.2%)	29 (19.7%)	2 (4.4%)	0.019
Off-pump	40 (20.8%)	28 (19.0%)	12 (26.7%)	0.297
General anaesthesia	143 (74.5%)	104 (70.7%)	39 (86.7%)	0.033
PVB analgesia	48 (25.0%)	42 (28.6%)	6 (13.3%)	0.048
Duration of intervention (min)	285 (240–370)	288 (240–366)	270 (235–400)	0.415
Duration of CPB (min)	110 (82–150)	110 (87–152)	100 (67–136)	0.135

Abbreviations: CPB, cardiopulmonary bypass; ECMO, extracorporeal membrane oxygenator; PVB, paravertebral block.

**Table 3 jcm-11-04901-t003:** Postoperative outcomes and major adverse events of overall (*n* = 192), INTERMACS Profile1–3 (*n* = 147) and Profile 4 (*n* = 45) patients.

Postoperative Outcomes	Overall (*n* = 192)	Profile 1–3 (*n* = 147)	Profile 4 (*n* = 45)	*p*
Intubation days	1 (1–3)	1 (1–3)	1 (1–2)	0.117
Tracheostomy	15 (7.8%)	13 (8.8%)	2 (4.4%)	0.527
Postop CVVH	65 (33.9%)	57 (38.8%)	8 (17.8%)	0.006
Days	9.0 (4.0–17.0)	10.0 (4.0–22.5)	6.5 (3.5–10.0)	0.039
RVF total	89 (46.3%)	75 (51.0%)	14 (31.1%)	0.026
Acute RVF	63 (34.4%)	54 (36.7%)	9 (20.0%)	0.046
Early-acute RVF	28 (14.6%)	27 (18.4%)	1 (2.2%)	0.007
Early-post-implant RVF	38 (19.8%)	32 (21.8%)	6 (13.3%)	0.563
Chronic RVF	30 (15.6%)	24 (16.3%)	6 (13.3%)	0.815
RVAD implantation	42 (21.9%)	39 (26.5%)	3 (6.7%)	0.015
Days	10.0 (6.0–13.0)	9.5 (6.0–13.0)	12.0 (3.0–12.0)	0.823
ECMO postop	13 (6.8%)	12 (8.2%)	1 (2.2%)	0.306
Days	3.0 (2.0–6.0)	3.0 (1.3–6.8)	4	0.954
NO inhalation	105 (54.7%)	82 (55.8%)	23 (51.1%)	0.609
**MajorAdverseEvents**	**Overall (*n* = 192)**	**Profile I–III (*n* = 147)**	**Profile IV (*n* = 45)**	** *p* **
Major bleeding	71 (37.0)	57 (38.8%)	14 (31.1%)	0.383
Revision for bleeding	60 (31.2%)	50 (34.0%)	10 (22.2%)	0.147
Cerebral event	47 (24.5%)	35 (23.8%)	12 (26.7%)	0.583
Non-fatal cerebral complications	28 (14.6%)	20 (13.6%)	8 (17.8%)	0.367
Ischemic	22 (11.5%)	17 (11.6)	5 (11.1%)	0.233
Haemorragic	6 (3.1%)	3 (2.0%)	3 (6.7%)	0.187
Fatal cerebral complications	19 (9.9%)	17 (11.6%)	2 (4.4%)	0.121
Bowel complications	26 (13.5%)	19 (12.9%)	7 (15.6%)	0.629
Fatal bowel complications	5 (2.6%)	5 (3.4%)	0 (0.0%)	0.593
Intra hospital documented infection	95 (49.5%)	80 (54.4%)	15 (33.3%)	0.017
VAD infection	2 (1.0%)	2 (1.4%)	0 (0.0%)	0.430
VAD thrombosis	15 (7.8%)	12 (8.2%)	3 (6.7%)	0.743
Thrombolysis	7 (3.6%)	7 (4.8%)	0 (0.0%)	0.203
Driveline infection total	37 (19.3%)	28 (19.0%)	9 (20.0%)	0.504
Within 6-months	14 (7.3%)	12 (8.2%)	2 (4.4%)	0.525
Over 6-months	23 (12.0%)	18 (12.2%)	5 (11.1%)	0.925
Postop-ICU (days)	8.0 (4.0–18.0)	10.0 (5.0–20.0)	5.0 (3.0–15.3)	0.010
In-hospital stay (days)	30.0 (20.0–45.0)	30.0 (20.5–52.5)	26.5 (20.0–41.5)	0.037
30-days death	25 (13.0%)	22 (15.0%)	3 (6.7%)	0.206
90-days death	43 (22.4%)	39 (26.5%)	4 (8.9%)	0.014
In-hospital mortality	48 (25.0%)	45 (30.6%)	3 (6.7%)	0.001
Cause of intra-hospital mortality				<0.001
Multiorgan failure	26	26 (17.7%)	0 (0.0%)	
Neurologic	12	10 (6.8%)	2 (4.4%)	
Bowel complications	3	2 (1.4%)	1 (2.2%	
Sepsis	7	7 (4.8%)	0 (0.0%)	
Rehospitalization	102 (53.1%)	76 (51.7%)	26 (57.8%)	0.648
Number of rehospitalization	2 (1–4)	2 (1–4)	2 (1–3)	0.996

Abbreviations: CVVH, continuous venovenous hemofiltration; ECMO, extracorporeal membrane oxygenation; ICU, intensive care unit; LVAD, left ventricular assist device; NO, nitrous oxide, RVAD, right ventricular assist device; RVF, right ventricular failure.

## Data Availability

The data that support the findings of this study are available from the corresponding author [T.B.] on request. The data are not publicly available due to restrictions because the information could compromise the privacy of research participants.

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
