# Peer review of "A Device Strategy-Matched Comparison Analysis among Different Intermacs Profiles: A Single Center Experience"

_jcm, 2022, doi:10.3390/jcm11164901_

Round 1
Reviewer 1 Report
The paper by Caraffa and co-workers reports their long-term experience with LVAD in patients stratified by INTERMACS classes.
The manuscript is interesting and well written and deserves publication following some minor revisions such as:
1) The main message this paper conveys is that the status of patients at the time of implant makes the difference in terms of results, especially in patients likely candidates to transplant. This means that a more “aggressive” indication to mechanical assistance should be considered. The message should be part of conclusions.
2) Figure 5 shows the cumulative risk of a patient being alive, dead or transplanted during the follow-up. This is a very interesting analysis and authors should elaborate a bit more on the inferences coming out of this analysis.
Please note Figure 5 is erroneously mentioned as Figure 3A in results.
3) Table 3
The way cerebral events are detailed in the table may result unclear to the reader. Some confusion may arise on the cause (ischaemic vs haemorrhage).
Author Response
Padova, 17th August 2022
EDITOR-IN-CHIEF
Journal Clinical Medicine
Dear Guest Editor (Piergiorgio Tozzi, MD)
We would be very thankful if you could consider for possible publication on “Journal Clinical Medicine” our paper entitled: “A DEVICE STRATEGY-MATCHED COMPARISON ANALYSIS AMONG DIFFERENT INTERMACS PROFILES: A SINGLE CENTRE EXPERIENCE” to be considered as Original Article.
- We replied point-by-point to all the kind comments made by the reviewers;
- We added new sentences
- Thanks to reviewers we improved the paper
Reviewer number 1
The paper by Caraffa and co-workers reports their long-term experience with LVAD in patients stratified by INTERMACS classes.
The manuscript is interesting and well written and deserves publication following some minor revisions such as:
Answer: We thank the reviewer for the kind comments
1) The main message this paper conveys is that the status of patients at the time of implant makes the difference in terms of results, especially in patients likely candidates to transplant. This means that a more “aggressive” indication to mechanical assistance should be considered. The message should be part of conclusions.
Answer: Perhaps introducing this concept into the conclusions is beyond the merits of this study. We agree with the reviewer that this is an important issue. Therefore, we added the following sentence on page 12 of the discussion: Our strategy allows in this subgroup group of patients to benefit from excellent survival and to be in optimal condition for transplantation (25).
2) Figure 5 shows the cumulative risk of a patient being alive, dead or transplanted during the follow-up. This is a very interesting analysis and authors should elaborate a bit more on the inferences coming out of this analysis. Please note Figure 5 is erroneously mentioned as Figure 3A in results.
Answer: We modified the figure mention. We added a new sentence within the discussion at page 12.
3) Table 3
The way cerebral events are detailed in the table may result unclear to the reader. Some confusion may arise on the cause (ischaemic vs haemorrhage).
Answer: We thank the reviewer for the comment. We have also added the row with hemorrhagic events in table 3 under ischemic brain events.
Reviewer number 2
I thank the authors for the interesting paper entitled: "A device strategy-matched comparison analysis among different intermacs profiles: a single center experience". The study was conducted on 192 patients undergoing LVAD implantation and the primary and secondary end-points were survival and major adverse events between profiles 1-3 vs profile 4.
The retrospective, observational, single-centric study concluded that: "LVAD implantation in elective patients was associated with better survival and lower complications incidence. LVAD implantation in BTC patients has to be considered before their conditions deteriorate and DT should be addressed to elective patients in order to guarantee acceptable results ".
The research question is clearly outlined. The study design is appropriate to answer the aim, the data is presented in a clear and appropriate way, and the conclusions answer the aims of the study.
The paper is of considerable interest as LVAD therapy is becoming the therapy of choice in ESHF patients in recent years and the INTERMACS profile represents a fundamental prognostic tool to guide clinical decision making. Furthermore, a correct indication and selection of patients, and a correct evaluation of the risks and benefits can actually represent a winning strategy.
I congratulate the authors as I believe that the paper can be of considerable help in clinical practice as the LVAD represents a valid alternative for critically ill patients in a shortage of donor era.
Answer : We thank the reviewer for the kind comments.
Reviewer 2 Report
I thank the authors for the interesting paper entitled: "A device strategy-matched comparison analysis among different intermacs profiles: a single center experience". The study was conducted on 192 patients undergoing LVAD implantation and the primary and secondary end-points were survival and major adverse events between profiles 1-3 vs profile 4.
The retrospective, observational, single-centric study concluded that: "LVAD implantation in elective patients was associated with better survival and lower complications incidence. LVAD implantation in BTC patients has to be considered before their conditions deteriorate and DT should be addressed to elective patients in order to guarantee acceptable results ".
The research question is clearly outlined. The study design is appropriate to answer the aim, the data is presented in a clear and appropriate way, and the conclusions answer the aims of the study.
The paper is of considerable interest as LVAD therapy is becoming the therapy of choice in ESHF patients in recent years and the INTERMACS profile represents a fundamental prognostic tool to guide clinical decision making. Furthermore, a correct indication and selection of patients, and a correct evaluation of the risks and benefits can actually represent a winning strategy.
I congratulate the authors as I believe that the paper can be of considerable help in clinical practice as the LVAD represents a valid alternative for critically ill patients in a shortage of donor era.
Author Response

(The authors gave the same response as above.)
